# A Preliminary Study Investigating the Impact of Musical Concerts on the Behavior of Captive Fiordland Penguins (*Eudyptes pachyrhynchus*) and Collared Peccaries (*Pecari tajacu*)

**DOI:** 10.3390/ani10112035

**Published:** 2020-11-04

**Authors:** Lara Fanning, Hannah Larsen, Peta S. Taylor

**Affiliations:** 1School of Environmental and Rural Science, Faculty of Science, Agriculture, Business and Law, University of New England, Armidale, NSW 2351, Australia; laraefanning@gmail.com; 2Wildlife Conservation and Science, Zoos Victoria, Melbourne, VIC 3052, Australia; hlarsen@zoo.org.au

**Keywords:** zoo animal welfare, animal behaviour, visitor effect, noise, animal management, music

## Abstract

**Simple Summary:**

Understanding the effects that human environments have on captive zoo animals is key when developing management procedures that foster good captive animal health and welfare. Through analysis of behavioural time budgets of Fiordland penguins and collared peccaries (solitary- and group-housed), we found that species’ behaviours and exhibit use altered when musical concerts were held at Melbourne Zoo. Fiordland penguins increased the use of a nest and pool and spent less time preening and interacting with the habitat on concert days compared to days when there was no concert. The solitary-housed peccary rested more and used the back and front of the exhibit more on concert days compared to days where there was no concert, while four group-housed peccaries rested more and were more vigilant on concert days. There were many time points when animal location and behaviour were unknown, which were related to concert days, highlighting the importance of monitoring the whole exhibit—or at least preferred habitats—when assessing potential stressors on behaviour. It is difficult to ascertain whether observed behavioural changes were indicative of stress or behavioural flexibility that allowed animals to cope; however, this research generates hypotheses for future investigations to better manage captive Fiordland penguins and collared peccaries on event days.

**Abstract:**

Captive animal welfare is important for establishments that exhibit species for education, conservation, and research. However, captive animals are often exposed to a number of potential stressors, such as visitors and anthropogenic noise. We aimed to identify the impact of a concert series on the behaviour of Fiordland penguins (*Eudyptes pachyrhynchus*; *n* = 2), and solitary- (*n* = 1) or group- (*n* = 4)-housed collared peccaries (*Pecari tajacu*). Animal behaviour, visitor density, and visitor behaviour was monitored pre-concert (afternoons; 16:00–19:00), during the concert (evenings; 19:00–21:00), and post-concert (nights; 21:00–00:00) on concert days (penguin *n* = 7 days; peccary *n* = 8 days) and in the same periods on days when there was no concert (penguin *n* = 8 days; peccary *n* = 6 days). Fiordland penguins spent more time surface swimming and diving in the pool on concert afternoons and evenings (all *p* < 0.001), more time in the nest on concert nights (*p* < 0.001), preened less on concert afternoons and nights (*p* = 0.019), and engaged with their habitat less on concert evenings and nights (*p* = 0.002) compared to these periods on days without a concert. The group-housed peccaries slept more in the afternoon and evening (*p* ≤ 0.01) and were more vigilant at night (*p* = 0.009) on concert days compared to no-concert days. The solitary-housed peccary slept more on concert nights (*p* = 0.035), rested more frequently across all time periods on concert days (*p* < 0.001), and used the front of the enclosure more across all concert time periods (*p* < 0.001) compared to no-concert days. We provide evidence that behaviour was altered on event days; however, we cannot determine the nature of these changes. Further research is needed to understand the impact of music concerts on zoo animal welfare.

## 1. Introduction

Animal welfare is a priority for modern zoos as maintaining a high standard of animal health and wellbeing enables zoos to meet animal conservation, education, research, and rehabilitation goals. Factors that can influence captive animal welfare include physiological variables such as health and nutrition and environmental variables such as exhibit design, social grouping, visitor presence, and abiotic stimuli. Developing evidence-based management practices allows welfare concerns to be identified and mitigated, thus facilitating improved welfare. In captive settings, it is often difficult to ascertain whether individual animals will adapt to changing anthropogenic environments. Stimuli present in zoo environments that can affect an animal’s welfare include general management practices such as animal handling, transportation, exhibit maintenance, zoo event days, and daily visitor presence [1]. While certain animal species display behavioral flexibility in responding to zoo stimuli and therefore may adapt to changing zoo environments, others may be more sensitive to such change, thus may be more challenging to manage [2]. Individual animals may respond differently to stimuli on the basis of prior experiences, physical condition, and a number of environmental variables, making zoo animal management particularly difficult [3]. Therefore, in order to promote positive welfare outcomes, zoos should adopt an evidence-based management framework in order to make informed decisions about management practices that rely on data from both the species and individual perspective [4]. 

One particular area of interest for zoo animal welfare management is understanding the effect of novel events on animal welfare. Many zoos host novel event and function days, such as concerts, fundraisers, and education days for zoo visitors. Event days often see an increase of visitor numbers and higher levels of ambient noise [1]. During such events, animals may be at risk of developing detrimental behavioral and physiological problems, particularly those that do not have social, environmental, and health needs met, or individuals maintained in enclosures that facilitate a high degree of visitor interaction [2,5,6,7]. A recent study by Bastian et al. (2020), which found that a majority of subject zoo-housed gorillas rested less during a late night zoo event compared to before the event and additionally noted changes in individual gorilla behavioural responses to the event, highlights the need to conduct studies on animal response to zoo event days [8]. Music has been reported to elicit both negative and positive responses in animals housed in both laboratory and zoo settings, depending on the species and individual [9,10,11]. 

The visitor effect is defined as changes in animal behaviour and physiology in response to visitor presence [12]. Studies have documented that high visitor density is correlated with increased aggression and stereotypies in gorillas [13]; increased urine cortisol in spider monkeys [14]; increased aggression, restless behaviour, and faecal cortisol concentrations in Indian blackbuck [15]; and increased avoidance, aggression, and vigilance in penguin species [16,17,18]. Contrastingly, Bloomfield et al. (2015) reported that orangutans re-positioned themselves to face visitor viewing windows when the windows were partially covered, which authors suggested may have indicated a preference for being closer to visitors or seeking visitor interaction [19].

Studies on captive animal welfare are rarely applicable across a wide range of species, and therefore many zoos now conduct focused species-specific or even individual-specific research. Due to the high degree of animal variation in response to both visitors and music, identifying at-risk individuals and mitigating the impact of visitors and noise may be key to maintaining good animal welfare on zoo event days and managing captive animals on the basis of scientific evidence.

Penguins are one of the most popular animal groups displayed in zoos [20]; however, there can be challenges associated with their housing. Two studies on little penguins found increased aggressive and avoidance behaviours, and increased glucocorticoid concentrations in response to visitor presence [16,17,18] in captive settings. However, Chiew et al. [17] detailed that captive little penguins’ responses to visitor presence was mitigated by regulating visitor proximity to two meters away from the exhibit fence [17]. Additionally, Ozella et al. (2015) noted that captive African penguins reduced pond use in response to high visitor densities following the opening of the exhibit, although this behavioural change was not observed two months into the study, suggesting habituation to visitor presence. It is also recognised that visitor behaviour, rather than visitor density or proximity to exhibits, can be the cause of behavioural changes [7]. 

However, other studies provide evidence that some penguin species, including gentoo (*Pygoscelis papua*) and Magellanic penguins (*Spheniscus magellanicus*), are highly adaptable and display improved biological functioning in captive settings [2,21]. Several studies have also outlined that captive penguin species display stable faecal corticosterone levels [22] and increased behavioural diversity [23] in response to visitor presence, suggesting that visitor presence may be perceived as a neutral or positive experience. 

The Fiordland penguin (*Eudyptes pachyrhynchus*) is native to isolated coastal areas of New Zealand and south-eastern Australia [24]. This species occurs in naturally low numbers in isolated wilderness areas; thus, Fiordland penguins are rarely exhibited in zoos, and studies surrounding Fiordland penguin welfare in captivity are limited. 

Contrastingly, a common species held across zoos is the collared peccary (*Pecari tajucu*), a medium-sized ungulate native to South America [25,26]. Collared peccaries are a popular zoo species given their small size, docile nature, and flexible diet and social groupings [27,28]. However, as a rainforest-dwelling prey species, the collared peccary has poor eyesight and is heavily reliant on hearing for survival [26]. This raises some concerns regarding captive collared peccary sensitivity to noise when kept in zoos, where ambient noise levels are higher than those that occur in their wild home ranges [7,29]. Additionally, a number of studies have outlined that wild peccaries can perceive humans as threatening and report that human presence can deter peccaries from feeding grounds and disturb circadian rhythms in the wild [30,31]. As the collared peccary is a prey species, captive individuals may perceive both visitors and general anthropogenic activity as aversive [32]. 

Using behavioural time sampling, this study investigated the behavioural responses of solitary-housed and group-housed collared peccaries, and two Fiordland penguins to an annual music concert series, the Twilights festival, held at Melbourne Zoo, Australia. During the seven-week Twilights concert series, Melbourne Zoo opened to visitors after regular opening hours and featured live musical performances each Friday and Saturday evening. This study aimed to identify the subject animals’ responses to visitor number and ambient noise levels. The findings of this research may aid zoo staff in managing these two species, which are poorly studied in captive settings [33], so that the best standard of animal care and welfare might be achieved. 

## 2. Materials and Methods 

This research was purely observational and did not involve any direct contact with animals or require any changes to normal zoo husbandry practices. This research was approved by the University of New England Human Ethic Committee (HE19–1000). 

### 2.1. Study Site

Melbourne Zoo is located in the Melbourne suburb of Parkville, approximately 4 km from Melbourne’s central business district. Over the study period, temperatures ranged from 9 °C to 38 °C, with the average temperature measuring 20 °C [34]. Rainfall on all study days was <2 mL, with the exception of 31st January, in which 6.6 mL of rain was reported by the Bureau of Meteorology [34].

### 2.2. The Melbourne Zoo Twilights Concert Series

The Twilights festival is a series of musical concerts held annually at Melbourne Zoo between January and March. The festival has run since 2008. The 2019 Twilights concert series ran from the 26 January to 9 March, with weekly concerts on Friday and Saturday evenings. Concerts began between 7:00 and 7:30 p.m. with music playing until 9:00 to 9:30 p.m. Visitors left the zoo between 9:00 and 9:30 p.m. and concert clean-up finished between 10:00 and 10:30 p.m. Weekly attendance rates fluctuated; however, the maximum guest capacity for each concert was 3000 people. Concert guests could access animal exhibits from 6:00 p.m. until 7:00 p.m., at which point security personnel directed visitors to the stage area (Figure 1) and restricted access to animal enclosures for the remainder of the evening. Concert audio equipment limited the music sound pressure levels to <98 dB at the sound desk directly in front of the stage. Musical genres played at the concert varied, but included blues rock, new wave, pop, rock, jazz, alternative, and classical genres. 

### 2.3. Study Animals and Exhibits

#### 2.3.1. Fiordland Penguins

The subject Fiordland penguins (*n* = 1 male; *n* = 1 female) were rescued on the beaches of the Mornington Peninsula (Victoria, Australia) in July 2018. The pair were rehabilitated and relocated to an outdoor exhibit at Melbourne Zoo, which also housed 25 little penguins. The two Fiordland penguins had not experienced a Twilights festival previously. 

The exhibit (332 m^2^) featured a large swimming pool with water flow, sandy vegetated areas, a water entrance platform with feeding stations and scales, and two den areas (Figure 2). One den area with enclosed nest boxes was allocated to the Fiordland penguins, while the other den area was allocated to the little penguins. Den sharing between species was prevented through the size and height of den entrances. Fiordland penguins had access to their own den site at all times. Visitors could view the penguin enclosure over a 1.2 m high wall from all points on the visitor path, while underwater viewing windows were located at the forefront of the exhibit pool area (Figure 2). 

#### 2.3.2. Peccaries

The solitary-housed peccary was a 22-year-old captive-born male. Due to aging and conspecific aggression, he was moved to a solitary exhibit space adjacent to the main peccary herd approximately one year prior to the 2019 Twilights concert series. The exhibit comprised two joined den areas with two sleeping boxes, as well as a large open grassed and vegetated area (Figure 2). The solitary peccary enclosure was not visible to visitors at any time.

The peccary group was comprised of two female and two castrated males between the ages of 7 and 12 years. All individuals were born at Melbourne Zoo and had never been relocated. The group peccary exhibit comprised a concrete den area with two timber nest boxes, and a large open grassed and vegetated area with a small wallowing pool (Figure 2). The group-housed peccaries were on display to zoo visitors. All peccaries had experienced between 7 and 11 previous Twilight festivals. 

### 2.4. Video and Acoustic Monitoring

Permanent motion sensor cameras (Axis P3225-LVE and P3707-PE, Axis Communications AB, Lund, Sweden) were installed in the penguin exhibit, and temporary motion sensory cameras (Swann 5 MP Super HD Thermal Sensing IP Bullet Cameras, Swann Communications Pty. Ltd., Melbourne, Australia) were installed in the peccary exhibit to continuously record animal behaviour and movement, as well as visitor attendance for the group-housed peccaries and penguins. Entire exhibits were not visible in camera scopes (Figure 3), and because the cameras functioned via motion detection, footage was unavailable when there was minimal animal movement or visitor presence. 

Class 2 sound level metres (Centre 323 Data Logger Sound Level Meter, Instrument Choice, Adelaide, Australia) were calibrated and then installed in the penguin and peccary exhibits (Figure 3). The peccary sound level metre was mounted atop the wall (>1 m high) separating the 2 peccary enclosures, and the penguin sound level metre was mounted on a stand (>1 m high) on the penguin entrance platform (Figure 3). Sound logger devices measured sound pressure levels (*LA*), measured in decibels (dB), between 30 and 130 dB, and 20 Hz to 8 kHz, across an A-weighted frequency spectrum every minute. Mean, minimum, and maximum sound pressure levels were then calculated for afternoons, evenings, and nights on concert and no-concert days by averaging the dB level over each respective 4-hour time period. The sound logger in the penguin enclosure was placed too close to running water, which resulted in a near-constant background noise of 73 dB. The data could therefore not be used with confidence and was excluded from the study.

### 2.5. Experimental Design

The study was conducted from 26 January 2019 through to March 9th, 2019. Behavioural sampling days (penguins *n* = 17 days; solitary peccary *n* = 14 days; group-housed peccaries *n* = 14 days) included two no-concert days (Wednesday and Thursday) and two concert days (Friday and Saturday) each week. Behavioural observations were recorded across three distinctive time periods during the afternoons (16:00*–*19:00), evenings (19:00*–*22:00), and nights (22:00*–*1:00) (Figure 4). Time periods were selected for the purpose of identifying what element of the concerts may have caused behavioural responses. Between 16:00 and 19:00, visitors were able to view animal exhibits and no music was played. Between 19:00 and 21:00, music was played but visitors were not able to view exhibits. After 22:00, all visitors left the zoo and music ceased to play. Observations were carried out by one trained observer. 

### 2.6. Behavioural Data Collection

Ethograms were developed for each species (Table 1 and Table 2*)* on the basis of two hours of initial footage observation and with reference to ethograms constructed by Sherwen et al. (2015) and De Faria et al. (2018). Animal location (Figure 3) and behaviour (Table 1 and Table 2) was recorded using instantaneous sampling for individual animals [35,36] or scan sampling for group housed animals [37] at 5 min intervals. Individual penguins were identifiable by differences in size, brow shape, gait, and beak shape; therefore, penguins were sampled as individuals. 

There were instances in which behavioural footage was missing, cameras were not activated, or the animals were not observable in the footage due to camera blind spots (Figure 3). These instances were recorded as such during sampling. 

### 2.7. Visitor Observations

Visitor number was recorded simultaneously with penguin and group-housed peccary observations. As the solitary-housed peccary was not on public display, we did not record visitor data for this individual. Visitors were counted if they were located at any location on the species’ respective visitor paths between points 1 and 2 (Figure 2).

### 2.8. Statistical Analysis

Statistical analysis was performed with SPSS statistical software (v. 23, IBM Crop, Armonk, NY, USA). The frequency of animal behaviours, location, and visitor number were summed hourly to calculate the percentage of time spent performing behaviours and time spent in locations. Due to a large quantity of data missing, any hour that was missing 6 or more data points (≥50%) was removed from the analysis (*n* = 23). Instances where cameras were not activated were removed for the solitary-housed peccary and penguins, however, were included for the group-housed peccaries. Following this process, a total of 1836 penguin, 1278 solitary-housed peccary, and 1512 group-housed peccary observations were analysed across 7 and 8 concert days for the penguins and peccaries, respectively, and 8 and 6 no-concert days for the penguins and peccaries, respectively.

A generalised linear model (GLM) with a Poisson log linear distribution was used to determine the difference in frequency of behaviours or location at each time point on concert and no-concert days. Treatment (concert or no-concert), time period (afternoon, evening, night), and the interaction between treatment and time period were included in all models as main effects, and week was included as a covariate. The penguin analysis also included penguin ID, independent and as two- and three-way interactions. Interactions that were not significant were removed from a model if removal improved the model fit, evidenced by the AIC value. Multiple comparisons were corrected with the Tukey method.

To determine if the number of visitors differed on concert days compared to no-concert days, we analysed three visitor periods: 4–5 p.m. no-concert days (regular zoo visitors on no-concert days), 4–5 p.m. concert days (regular zoo visitors on concert days), and 6–7 p.m. concert days (after-hours concert visitors on concert days) with a one-way analysis of variance (ANOVA), followed by Tukey’s post hoc analysis.

An independent sample *t*-test was used to determine whether minimum, maximum, or average ambient sound pressure levels differed between concert and no-concert days. A scale response generalised linear model with Tukey’s post hoc analysis was utilised to test the effect of treatment and time period (afternoons, evenings, nights) and the interaction between treatment and time period on minimum, maximum, and average ambient sound pressure levels. 

## 3. Results

### 3.1. Fiordland Penguins

#### 3.1.1. Visitor Number at Penguin Exhibit

The average number of visitors per hour between 4 and 5 p.m. at the penguin enclosure did not differ between no-concert days and concert days (*p* = 0.133). On concert days, there were more visitors at the penguin enclosure on average between 4 and 5 p.m. (86.1 ± 25.8 visitors) compared to 6 and 7 p.m. on concert days (35.0 ± 8.4 visitors) and 4 and 5 p.m. on no concert days (52.7 ± 12.7 visitors). 

#### 3.1.2. Impact of Concert on Location and Behaviour

Penguins used the entrance platform (ꭓ^2^_(1, 66)_ = 23.9, *p* < 0.001) more frequently in the night compared to afternoon and evening, and this effect was greater on no-concert days (Table 3). Penguins spent more time in the pool (ꭓ^2^_(1, 66)_ = 30.3, *p* < 0.001) and in the nest (ꭓ^2^_(1, 66)_ = 57.0, *p* < 0.001) on concert days compared to no-concert days (Table 3). Penguins spent more time surface swimming (ꭓ^2^_(2, 74)_ = 18.0, *p* < 0.0001) and diving (ꭓ^2^_(2, 74)_ = 21.4, *p* < 0.0001) and less time engaging with the habitat (ꭓ^2^_(1, 66)_ = 12.7, *p* = 0.002) and preening (ꭓ^2^_(2, 74)_ = 5.5, *p* = 0.019) on concert days compared to no-concert days (Table 3). 

#### 3.1.3. Individual Differences

The female penguin used the nest more frequently than the male on no-concert days (ꭓ^2^_(1, 66)_ = 11.9, *p* = 0.001). Overall, the male penguin spent more time in the vegetated patch (ꭓ^2^_(1, 66)_ = 23.0, *p* < 0.001) and a greater proportion of time surface swimming (ꭓ^2^_(1, 74)_ = 4.8, *p* = 0.029) than the female penguin. The female penguin spent a lower proportion of time resting than male in the evening and night, but not during the afternoon (ID × time interaction: ꭓ^2^_(2, 66)_ = 12.0, *p* = 0.003). The female spent a greater proportion of time engaging with the habitat in the evening and night compared to the male at night (ꭓ^2^_(1, 66)_ = 3.9, *p* = 0.028). 

#### 3.1.4. Unknowns

There was a high frequency of unknown behaviour and locations (Table 3). These values were included in the analysis to avoid any potential bias, for example if penguins were less conspicuous related to treatment. Indeed, penguins were less visible on concert days compared to no-concert days (ꭓ^2^_(1, 74)_ = 31.94, *p* < 0.001; Table 3) and there was a greater proportion of unknown behaviours during the evening and night on concert days compared to no-concert days (ꭓ^2^_(1, 66)_ = 12.5, *p* = 0.002; Table 3).

### 3.2. Sound Pressure Levels at the Peccary Exhibits

Average, minimum, and maximum sound pressure levels at the peccary exhibit were greater in the afternoon and evenings on concert days compared to all time points on no-concert days (mean F_(1162)_ = 46.1, *p* < 0.001; minimum F_(1162)_ = 25.9, *p* < 0.00; maximum F_(1162)_ = 34.4, *p* < 0.001; Table 4).

### 3.3. Solitary Peccary

#### 3.3.1. Impact of Concert on Location and Behaviour

There was no interaction between treatment and time on any behaviour or time spent in any location (all *p* > 0.05; Table 5). The peccary was located at the front (χ^2^_(1, 36)_ = 27.9, *p* < 0.001) and back (χ^2^_(1, 36)_ = 6.4, *p* = 0.011) of the exhibit more frequently on concert days compared to no-concert days (Table 5). The peccary rested (χ^2^_(1, 36)_ = 7.8, *p* < 0.001) and slept (χ^2^_(1, 36)_ = 4.5, *p* = 0.035) more frequently on concert days compared to no-concert days (Table 5). Vigilance, locomotion, and standing behaviours were rare (Table 5).

#### 3.3.2. Unknowns

Missing footage accounted for a large portion of the dataset. Missing footage was more prevalent in the afternoon on no-concert days and was at the lowest level during the evening on concert days (treatment × time interaction χ^2^_(2, 36)_ = 13.6, *p* = 0.001; Table 5). The peccary was less visible during the evening on both concert and no-concert nights, and more visible during the night on concert days compared to no-concert days (treatment × time interaction χ^2^_(2, 36)_ = 26.6, *p* < 0.001; Table 5). Cameras were inactive more frequently in the evening on concert days compared to the evening on no-concert days (treatment × time interaction χ^2^_(2, 36)_ = 14.9, *p* = 0.001; Table 5).

### 3.4. Peccary Group

#### 3.4.1. Visitor Number at Group-Housed Peccary Exhibit

There were few visitors per hour to the group peccary exhibit and no difference in the mean number of visitors between 4 and 5 p.m. on concert and no-concert days or between 4 and 5 p.m., 6 and 7 p.m., and 7 and 10 p.m. on concert days (visitors/h: 4*–*5 p.m. no-concert days 1.0 ± 1.0; 4*–*5 p.m. concert days 1.3 ± 1.1; 6*–*7 p.m. concert days 3.8 ± 1.1; 7*–*10 p.m. concert days 2.7 ± 1.0; all *p* > 0.05). 

#### 3.4.2. Impact of Concert on Location and Behaviour

Use of the den was less frequent during afternoons on no-concert days and most frequent during the evening on no-concert days (χ^2^_(2,36)_ = 10.1, *p* = 0.007; Table 6). Use of the back of the exhibit was most frequent during the evening on concert days (χ^2^_(2,36)_ = 6.4, *p* = 0.012; Table 6). Use of the box (χ^2^_(1, 36)_ = 9.9, *p* = 0.002) and front of the exhibit (χ^2^_(1, 36)_ = 4.6, *p* = 0.032) was more frequent on concert days compared to no-concert days (Table 6). 

The peccaries were engaged with the habitat less frequently during the night on no-concert days compared to concert days when peccaries were less interactive with the habitat during the afternoon (treatment × time period interaction χ^2^_(2,36)_ = 11.9, *p* = 0.003; Table 6). The peccaries rested less frequently during the afternoon on no-concert days and more at night on concert days (treatment × time period interaction χ^2^_(2, 36)_ = 8.3, *p* = 0.016; Table 6). Additionally, the peccaries slept less frequently during the afternoon and evening on no-concert days compared to concert days (treatment × time period interaction χ^2^_(2, 36)_ = 20.0, *p* < 0.001; Table 6). Peccaries were more vigilant during the evening compared to afternoon and nights, and more vigilant during concert nights compared to no-concert nights (treatment × time period interaction χ^2^_(2,36)_ = 6.7, *p* = 0.009; Table 6).

#### 3.4.3. Unknowns

The cameras were more inactive during the evening and afternoon on concert days compared these periods on no-concert days, and more inactive on concert nights compared to no-concert nights (χ^2^_(2,36)_ = 12.5, *p* = 0.002; Table 6). There was more unknown behaviour during the afternoon on concert days compared to no-concert days (χ^2^_(2, 36)_ = 21.7, *p* < 0.001).

## 4. Discussion 

This preliminary study was unable to determine what, if any, welfare impacts the concert series had on the individual animals, or to determine specific causation of the behavioural changes observed. It nonetheless provides useful information with regard to behavioural patterns and habitat use during events that can be useful in making informed management decisions.

### 4.1. Fiordland Penguins

This study demonstrated that both the individual Fiordland penguins altered behaviour and exhibit use in response to a music concert series. The two Fiordland penguins used the pool and nest more on concert days compared to no-concert days. The penguins preened and interacted with the habitat less frequently on concert days and the female penguin displayed more locomotive behaviours on concert afternoons and evenings. A high degree of behavioural variation was noted between the individual penguins, and this was related to concert conditions. The male penguin spent more time resting and in the vegetated patch and less time at the water entrance platform on concert days compared to the female penguin. The female penguin used the nest box more on no-concert days. 

There was no difference in the number of visitors at the penguin exhibit in the last hour of zoo opening times on concert and no-concert days. However, there were significantly fewer visitors before the concert (6*–*7 p.m.) compared to regular opening hours (4*–*5 p.m.). Chiew et al. (2019) reported that captive little penguins increased their distance from visitor viewing windows when visitors were present, suggesting that this was a visitor avoidance response associated with fear or frustration. Similar responses were outlined in little penguins by Sherwen et al. (2015) and African penguins by Ozella et al. (2015). We found little evidence that visitors caused a fear or flight response in the Fiordland penguins, as evidenced by no increased use of the out-of-sight nest area during visitor periods. In the current study, the Fiordland penguins were observed using the pool more on concert afternoons, compared to concert evenings, when visitor numbers were higher. This could be a related to the penguins’ curiosity of visitors, or their seeking out of visitor interaction when visitor numbers were higher. Sound travels poorly from air to water [38]; as such, the pool may have minimised disturbances associated with visitors and musical noise for the Fiordland penguins. However, increased pool use was observed on concert days before the music started, indicating the penguins did not utilise this space as a retreat from concert-related noise. However, it is also possible that the penguins were seeking refuge in the water from ambient noise created by increased visitor numbers, a trend noted in other studies on captive aquatic species exposed to high visitor numbers [18,39]. Further investigation is required to understand the animal–visitor relationship for Fiordland penguins.

### 4.2. Peccaries

The solitary and group-housed peccaries showed similar behavioural responses to the concerts: increased resting and sleeping on concert days, compared to no-concert days. Additionally, the group-housed peccaries were more vigilant in the night on concert days, compared to the same time on no-concert days. Thus, the musical concert series appeared to cause behavioural changes in the species, regardless of social group. However, it is important to note that specific causational relationships associated with the concert series are not able to be determined, and other factors, such as weather, weekend visitation, husbandry routines, and individual condition may have also influenced the observed behavioural changes.

On the basis of the sound pressure data recorded at the peccary exhibit, we found that concert days in general were significantly louder than no-concert days overall (including outside of music times). Specifically, afternoons on concert days were approximately 10 dB louder than afternoons on no-concert days, suggesting that visitors were noisier on concert days prior to concerts beginning. On the basis of average sound pressure levels measuring 73 dB during concerts, the concert series has a potential to pose a welfare risk to animals. Similar construction-related sound pressure levels were described by Orban et al. (2017), and these may have contributed to an increase in the proportion of negative welfare indicators recorded for the subject giant anteater. Additionally, Pelletier et al. (2020) outlined that the sound pressure levels within an urban zoo were higher in particular areas, especially indoor or contained areas, and suggested that higher noise levels could cause damage to the mammalian ear structures [40], which raises some concerns for the peccaries that were exposed to prolonged and high sound pressure levels. 

Despite ambient noise levels being higher during concerts, very few behavioural changes indicated that any of the animals experienced a stress response. This might be due to the environmental features of the exhibit (e.g., shrubs/dens) providing sound buffers and reducing the impact of auditory stress on the animals. Of note, foliage would not provide as significant of a sound buffering effect when compared to more formidable objects such as walls of a den or larger and more solid objects within the exhibit.

The group-housed peccaries, who had direct line of sight to visitors and the concert stage, used the out-of-sight box and den more frequently on concert day afternoons, although we cannot determine whether this was related to higher numbers of regular zoo visitors, concert set-up activity, after-hour zoo visitors, or increased ambient noise. The frequency of habitat engagement and vigilance by the group-housed peccaries was greater during the night (post-concerts) of concert days compared to the night of no-concert days. However, increased habitat engagement would likely cause a natural incline of vigilance behaviours [41], and thus it is difficult to determine whether heightened vigilance was a result of fear and alarm, or a natural behavioural response associated with an increase in active behaviours. 

The group-housed peccaries slept more and used the den location more in the afternoon on concert days compared to no-concert days. Changes to resting patterns are difficult to interpret; inactivity has been linked to animal boredom, hiding or freezing, pain, ill health, lethargy, and depression [2,42,43,44]. Contrastingly, increased inactivity can also be considered a sign of good welfare, and may indicate that an animal is relaxed or content [43]. There were no stereotypic or abnormal behaviours observed during the study that would indicate the peccaries were experiencing boredom or stress-induced inactivity [45]. Although it is not possible to determine the causality of increased den use, the peccaries may have utilised the den to avoid disturbance related to concert set-up, due to the den potentially being perceived as a safe refuge zone. As the den is designed as a sleeping area, it is unlikely to support many other behavioural functions; thus, if the peccaries used this space as a retreat, it is then reasonable to infer sleep would also increase. 

Furthermore, increased frequency of resting in the evenings and nights on concert days was negatively related to the frequency of missing footage and the inability to locate the solitary peccary. As such, increased resting on concert days may have occurred on no-concert days but simply could not be recorded as such. 

Future research into the animal–visitor relationship for the group-housed peccaries is needed. Experimental research designs that control for visitor number, behaviour, music type, and sound pressure levels will provide better opportunity for interpreting results and making informed decisions that improve welfare outcomes for the individual animals. 

### 4.3. Future Evidence-Based Management Considerations

It is well documented that an animals’ response to stressors, especially noise, varies vastly between individuals [46]. Additionally, an animal’s response to music is influenced by the animal’s sex, age, physical condition, social grouping, and prior experience [1,46], and thus it is difficult to infer what caused the behavioural changes between the two individual penguins, the solitary peccary, and the peccary group. However, these results highlight the importance of monitoring independent captive animals and tailoring management practices to an individual’s needs with the aim of achieving the best welfare outcome on the basis of scientific inquisition. 

Missing data caused by limited remote monitoring capabilities*—*both lack of cameras in key locations and lack of motion activated recordings*—*highlights a problem that is common in zoo-based research and a limiting factor for effective evidence-based management. However, the alternative method of live observations on animals may also influence animal behaviour, particularly during hours when visitor access is restricted, and can be limited due to a lack of appropriate vantage points. Both the visibility and the potential influence on animal’s behaviour need to be considered when determining the best observation method when collecting data for evidence-based management projects. 

## 5. Conclusions

While the findings of this study were somewhat inhibited by a large portion of unknown data, the results are, nevertheless, important for providing zoo staff with detailed information about individual behavioural responses to music concerts and improving remote video monitoring in animal exhibits. Several findings may aid in developing tailored management practices that can improve individual animal care and welfare as well as strategies for improving evidence-based management capabilities. 

Evidence suggests that all three of the studied groups experienced behavioural changes. However, for all three study groups, there was not enough evidence to determine whether concert conditions incited stress-induced behavioural responses, and thus a net-negative welfare outcome. Exhibit design, through provision of boxes and sheltered out-of-sight areas, may have allowed the animals to control their environments and cope with changing visiting hours, visitor numbers, and noise disturbance, resulting in a net-neutral welfare outcome. This study provides valuable information regarding the importance of functional zones within exhibits and adequate video monitoring facilities for examining animal health and welfare, as well as highlighting the necessity of performing research on independent animals in zoos, given individuals respond differently to environmental changes. The results also highlight areas for future research, which may accurately identify which elements, if any, of concert conditions can trigger behavioural responses, and whether these behavioural responses indicate a specific welfare outcome. 

## Figures and Tables

**Figure 1 animals-10-02035-f001:**
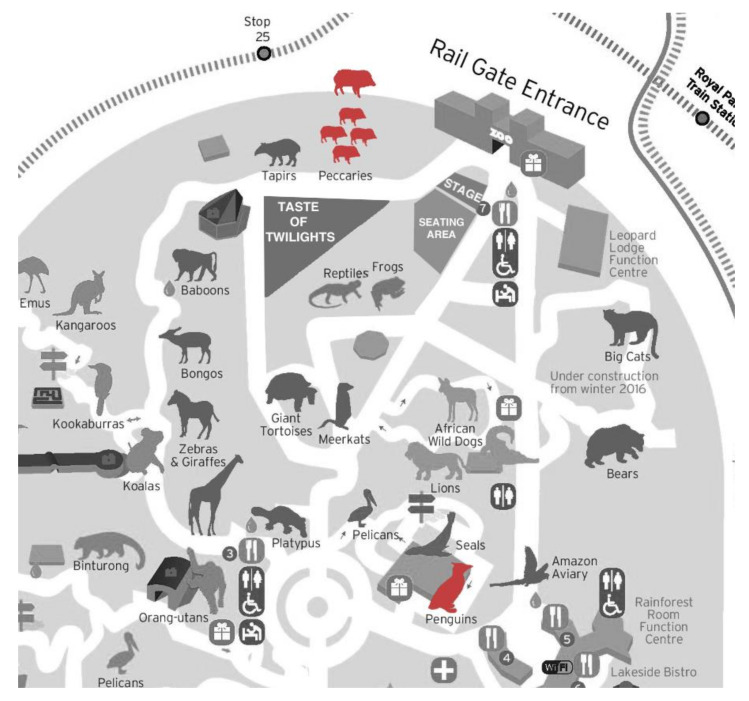
Map of Melbourne Zoo with location of subject species highlighted in red. The group peccary exhibit was located approximately 50 m from the Zoo Twilights Stage. The solitary peccary exhibit was directly behind the group-housed peccary exhibit, approximately 75 m from the Zoo Twilights Stage. The penguin exhibit was located approximately 300 m from the Zoo Twilights Stage. The Taste of Twilights section indicates an area where food could be purchased. Map adapted from “Melbourne Zoo Map” by Zoos Victoria, 2019, http://ontheworldmap.com/australia/city/melbourne/melbourne-zoo-map.html. Copyright 2019 by Zoos Victoria.

**Figure 2 animals-10-02035-f002:**
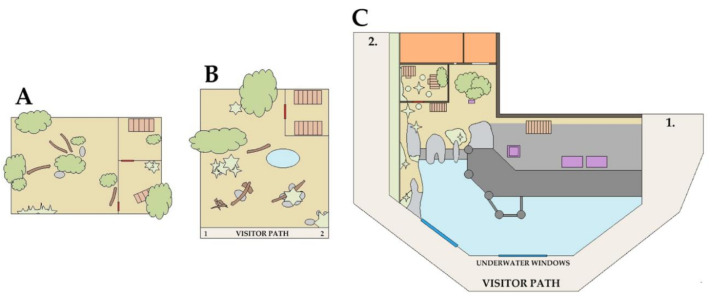
Diagram of the solitary peccary (**A**), group peccary (**B**), and penguin (**C**) exhibits at Melbourne Zoo, including sand or dirt (beige), water (blue), low vegetation (<1 m) and grass (light green), trees (>1 m) (dark green), boxes (light slatted brown), logs (brown), stones (light grey), concrete (mid-grey), timber water entrance platform (dark grey), feeding boxes or scales (purple), and out of exhibit areas (orange) and visitor paths (labelled).

**Figure 3 animals-10-02035-f003:**
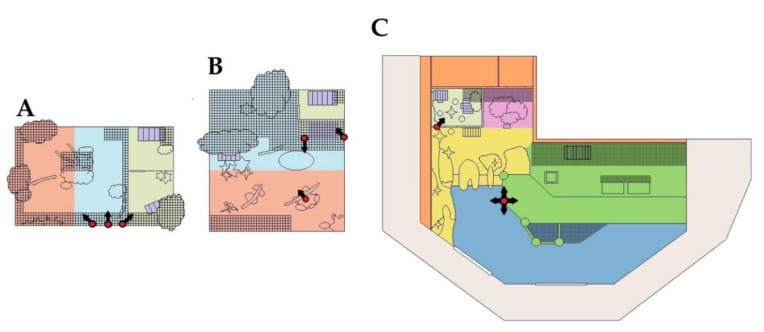
Study zones and camera locations and orientation in the (**A**) solitary-housed and (**B**) group-housed peccary exhibits—including front of exhibit (orange), back of the exhibit (blue), den (green), and boxes (purple), and (**C**) penguin exhibit—including dens (light green), boxes (purple), vegetated patch (pink), shoreline (yellow), water entrance platform (dark green), and out of exhibit area (orange). Red circles with arrows indicate the location and direction of the cameras. Gridded zones indicate locations outside the field of view of any camera.

**Figure 4 animals-10-02035-f004:**
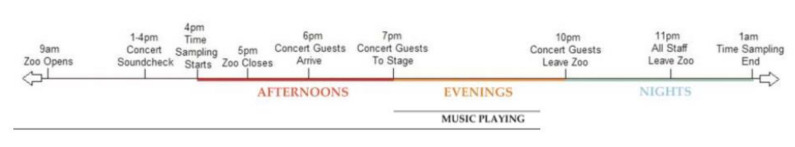
Timeline of events relating to penguin behavioural data collected on concert and no-concert days throughout afternoon, evening, and night periods.

**Table 1 animals-10-02035-t001:** Behavioural ethogram for Fiordland Penguins at Melbourne Zoo.

Category	Behaviour	Description
*Locomotion*	Walk	Slow movement in any direction while animal stands in an upright position. Head and chest may be hunched.
	Run	Fast movement in any direction while animal stands in an upright position. Head and chest may be hunched.
*Surface swim*	Surface swim	Floating or locomotion on surface of water.
*Dive*	Dive	Locomotion whilst completely submerged under water.
*Vigilance*	Vigilance	Stands in erect position with neck stretched above the shoulders and eyes open. Head and eyes may be orientated towards one point or searching for disturbance.
*Rest*	Rest	Lies on belly with head placed against the ground or supported on chests OR stands upright with head resting on neck/shoulders and flippers tucked to body.
	Stand	Remains stationary with two feet on the ground for at least 5 s, but body, head, and limbs may move. Head held upright (not propped against chest).
*Habitat engagement*	Exploration	Penguin examines item or environmental feature for at least 3 s, orientating eyes and head towards point of interest and often touching with beak. May be stationary or walking.
	Forage	Using beak, penguin picks up food item or nesting substrate including sticks, rocks, twigs, and leaves.
*Preen*	Preen	Uses beak to peck, stroke, or comb feathers in any region of the body. Animal either stationary on land or standing in or floating on water.
*Human interaction*	Keeper interaction	Orientates body and eyes to keeper; moves towards keeper or is in gentle contact with keeper. Excludes *antagonistic* behaviours towards keeper.
	Visitor interaction	Tapping visitor window glass with beak when visitor is present, or following visitor in the water or on the land, or orientating head and eyes to watch visitor through glass screen or over enclosure wall.
*Cohabitant interaction*	Affiliative	Using beak, pecks, strokes, or combs feathers of conspecific; standing beside or before another penguin and bowing the head towards it; touching beaks and/or the head with a conspecific; all mating behaviours.
	Antagonistic	Uses the beak to peck at another penguin, resulting in other penguin moving away or showing aggression; pursuing another penguin at a run with head lowered below shoulders, and eyes and head orientated towards target being chased; penguin faces another penguin with head held at same level or higher, and stares at other bird for at least two seconds; penguin stands with feathers raised and neck outstretched while conspecific lowers head.

**Table 2 animals-10-02035-t002:** Ethogram for collared peccaries at Melbourne Zoo.

Category	Behaviour	Description
*Locomotion*	Walk	Locomotion in any direction. At least two hooves remain on ground.
*Rest*	Rest	Lies either with belly or side of body touching the ground. Eyes may be opened or closed; however, peccary remains responsive to small environmental changes.
*Sleep*	Sleep	Lies either with belly or side of body touching the ground. Eyes are closed. Unresponsive to small changes in environment.
*Stand*	Stand	Remains stationary with all four hooves on the ground and legs straight. Head held level with back or below back.
*Habitat engagement*	Forage	Snout disturbs grass, vegetation, substrate, or water in a continuous “searching” motion. Snout may be used to root ground.
	Eat	Navigation of food item into mouth.
	Sniff	Snout touches and moves across a substrate. Head may also be lifted into the air while snout twitches or continues to move.
	Wallow	Substrate is manipulated by head, body, or legs so that peccary’s head, body, or legs are partially or entirely coated in substrate. May be performed on land or in water, while standing, sitting, or lying.
	Enrichment use	Uses any part of the body to touch enrichment source.
*Vigilance*	Vigilance	Head held level with or above the shoulders, ears pointed towards one point and eyes focused on same point. Animal may be standing or lying.

**Table 3 animals-10-02035-t003:** Mean ± SEM frequency of behaviours (% total observations) and locations of Fiordland penguins on concert days or days without a musical concert (Trt) across three time periods (afternoon, 4 to 6 p.m.; evening, 7 to 10 p.m.; night, 10 p.m. to 1 a.m.). Two penguins were observed and were included (ID) in each statistical model.

	No Concert			Concert						Trt × Time	Trt × ID	Time × ID	Trt × Time × ID
	Afternoon	Evening	Night	Afternoon	Evening	Night	Trt	Time	ID				
***Location***													
Vegetated patch	5.3 ± 2.2	8.3 ± 2.8	0.0 ± 0.0	7.9 ± 3.1	6.9 ± 3.9	0.0 ± 0.0	0.114	0.351	<0.001	0.033	0.005	0.927	0.996
Nest	0.7 ± 0.5	10.2 ± 6.1	25.5 ± 9.8	0.0 ± 0.0	13.5 ± 6.2	66.7 ± 21.1	<0.001	<0.001	<0.001	0.096	0.001	0.588	0.257
Entrance platform	11.8 ± 1.7	1.4 ± 0.5	25.0 ± 16.4	6.9 ± 2.0	5.1 ±1.2	5.6 ± 4.1	0.056	<0.001	0.939	<0.001	0.423	0.818	0.769
Shoreline	8.7 ± 2.2	12.7 ± 2.6	0.0 ± 0.0	4.8 ± 0.8	9.3 ± 2.2	0.0 ± 0.0	<0.001	1.000	0.186	1.000	0.102	0.867	0.905
Pool	15.5 ± 3.4	8.3 ± 1.5	0.0 ± 0.0	29.4 ± 3.2	20.7 ± 3.1	0.0 ± 0.0	<0.001	<0.001	0.271	0.711	0.617	0.828	0.835
Den	3.0 ± 1.1	7.7 ± 2.5	25.5 ± 9.8	2.0 ± 1.3	5.3 ± 1.4	0.0 ± 0.0	0.155	<0.001	0.436	0.818	0.683	0.822	0.611
Unknown location	18.9 ± 1.2	12.9 ± 1.7	3.4 ± 1.7	17.6 ± 1.5	10.2 ± 2.2	3.3 ± 2.1	0.048	<0.001	0.139	0.037	0.008	0.492	0.125
***Behaviour***													
Locomotion	4.8 ± 1.2	4.3 ± 1.1	3.1 ± 3.1	4.4 ± 0.9	3.9 ±1.5	0.0 ± 0.0	0.917	0.002	0.070	0.555	0.831	0.725	0.036
Surface swimming	6.4 ± 1.4	2.8 ± 0.9	0.0 ± 0.0	12.9 ± 1.5	8.4 ± 2.4	0.0 ± 0.0	<0.001	<0.001	0.029	0.826	0.292	0.683	0.486
Diving	3.8 ± 1.3	3.1 ± 0.7	0.0 ± 0.0	12.5 ± 1.3	4.7 ± 1.4	0.0 ± 0.0	<0.001	<0.001	0.266	0.182	0.568	0.594	0.741
Vigilance	6.3 ± 1.3	4.3 ± 1.6	1.6 ± 1.1	4.2 ± 1.0	7.8 ± 1.8	1.4 ± 1.4	0.986	0.001	0.763	0.198	0.222	0.632	0.097
Resting	9.8 ± 2.7	9.5 ± 2.0	18.8 ± 11.2	6.0 ± 2.0	4.9 ± 2.0	4.2 ± 2.8	<0.001	0.001	<0.001	0.024	0.561	0.003	0.690
Habitat engagement	1.0 ± 0.6	4.9 ± 2.5	5.2 ± 5.2	0.4 ± 0.3	0.6 ± 0.4	0.0 ± 0.0	<0.001	0.002	0.278	0.808	0.106	0.028	-
Preening	7.2 ± 0.0	4.6 ± 1.4	4.2 ± 2.7	3.8 ± 1.1	4.7 ± 1.2	0.0 ± 0.0	0.019	0.001	0.275	0.303	0.874	0.433	0.837
Human interaction	0.0 ± 0.0	0.0 ± 0.00	0.0 ± 0.0	2.2 ± 0.8	0.0 ± 0.0	0.0 ± 0.0	-						
Cohabitant interaction	1.0 ± 0.3	3.0 ± 0.0	0.0 ± 0.0	1.4 ± 0.5	5.0 ± 1.4	0.0 ± 0.0	0.370	0.041	0.873	0.996	0.882	0.821	0.833
***Unknown***													
Unknown behaviour	4.7 ± 1.2	11.0 ± 16.4	46.4 ± 16.4	3.9 ± 1.6	17.3 ± 6.7	66.7 ± 21.1	0.001	<0.001	0.660	0.002	<0.001	<0.001	0.521
Footage missing	2.4 ± 2.4	2.6 ± 0.5	0.5 ± 0.5	0.0 ± 0.0	0.0 ± 0.0	0.0 ± 0.0	-						
Animal not visible	15.5 ± 2.0	3.1 ± 0.5	0.5 ± 0.5	28.0 ± 3.2	9.1 ± 1.6	1.4 ± 1.4	<0.001	<0.001	0.587	0.418	0.868	0.475	0.854
Camera inactive	37.1± 4.4	41.8 ± 9.7	19.8 ± 9.7	20.4 ± 3.1	26.6 ± 4.8	26.4 ± 17.0	0.001	<0.001	0.588	0.195	<0.001	0.002	0.266

- indicates that the behaviour was too rare for analysis.

**Table 4 animals-10-02035-t004:** Average sound pressure levels (*LA*) ± SEM measured at the peccary exhibits on treatment days (concert or no-concert days) and across three time periods (afternoon, 4 to 6 p.m.; evening, 7 to 10 p.m.; night, 10 p.m. to 1 a.m.).

		No Concert			Concert	
Sound Pressure Level	Afternoon	Evening	Night	Afternoon	Evening	Night
*LA* _avg_	50.1 ± 2.0 _a_	48.0 ± 0.7 _a_	48.7 ± 1.8 _a_	61.1 ± 1.3 _b_	72.8 ± 1.7 _c_	48.0 ± 1.6 _a_
*LA* _min_	44.7 ± 0.3 _a_	43.3 ± 0.5 _a_	41.8 ± 0.8 _a_	48.9 ± 1.2 _b_	54.1 ± 1.4 _c_	42.4 ± 0.4 _a_
*LA* _max_	65.0 ± 3.8 _a_	57.3 ± 1.4 _ad_	61.3 ± 2.2 _a d_	82.2 ± 1.67 _b_	87.5 ± 1.2 _c_	55.9 ± 2.5 _d_

Values presented in dB. Different subscripted letters indicate significant differences between the interaction of treatment and time (*p* < 0.05).

**Table 5 animals-10-02035-t005:** Mean ± SEM proportions of behaviours (% total observations) and location of a solitary-housed collared peccary on concert or no-concert days (Trt) across three time periods (afternoon, 4 to 6 p.m.; evening, 7 to 10 p.m.; night, 10 p.m. to 1 a.m.).

	No Concert			Concert					
	Afternoon	Evening	Night	Afternoon	Evening	Night	Trt	Time	Trt × Time
***Location***									
Den	0.0 ± 0.0	2.3 ± 0.9	0.0 ± 0.0	0.0 ± 0.0	4.1 ± 1.7	1.4 ± 0.7	0.087	0.009	-
Exhibit front	7.9 ± 4.1	16.2 ± 6.5	12.5 ± 2.9	15.5 ± 2.9	31.6 ± 7.5	32.3 ± 8.7	<0.001	<0.001	0.498
Exhibit back	2.8 ± 2.3	11.5 ± 6.5	30.4 ± 11.2	20.4 ± 11.1	6.3 ± 5.2	13.2 ± 7.1	0.011	<0.001	0.715
***Behaviour***									
Locomotion	0.0 ± 0.0	7.4 ± 2.3	3.2 ± 1.8	0.3 ± 0.3	5.6 ± 2.1	2.8 ± 1.0	0.481	0.002	0.831
Habitat engagement	7.4 ± 3.6	22.7 ± 7.6	12.5 ± 3.6	9.4 ± 4.8	24.7 ± 6.3	8.3 ± 3.7	0.851	<0.001	0.241
Rest	0.0 ± 0.0	0.0 ± 0.0	1.9 ± 1.4	5.2 ± 4.8	11.1 ± 5.8	8.3 ± 2.2	<0.001	0.048	-
Sleep	3.2 ± 0.0	0.0 ± 0.0	14.7 ± 10.8	0.0 ± 0.0	0.0 ± 0.0	26.7 ± 7.7	0.035	<0.001	-
Vigilance	0.0 ± 0.0	0.0 ± 0.0	0.0 ± 0.0	0.0 ± 0.0	0.7 ± 0.7	0.7 ± 0.5	-		
Stand	0.0 ± 0.0	0.0 ± 0.0	0.5 ± 0.5	0.0 ± 0.0	0.0 ± 0.0	0.0 ± 0.0	-		
***Unknown***									
Footage missing	60.2 ± 20.0 _a_	16.7 ± 8.6 _b_	32.7 ± 0.5 _c_	54.2 ± 15.1 _a_	4.5 ± 3.8 _d_	31.9 ± 0.5 _c_	<0.001	<0.001	0.001
Animal not visible	9.7 ± 6.6 _a_	40.7 ± 6.1 _b_	28.5 ± 8.1 _b_	17.4 ± 5.7 _abc_	27.4 ± 6.7 _b_	8.0 ± 2.7 _ac_	<0.001	<0.001	<0.001
Camera inactive	19.4 ± 12.1 _a_	12.5 ± 5.5 _bc_	6.1 ± 3.2 _c_	13.5 ± 6.8 _ab_	26.0 ± 11.5 _ad_	13.2 ± 8.6 _abc_	0.010	0.001	0.001

Different subscript letters denote significant changes between treatment days and time periods (*p* < 0.05). - indicates that no analysis was performed as the behaviour or location was too rare.

**Table 6 animals-10-02035-t006:** Mean proportion ± SEM of behaviour and location of group-housed collared peccaries on concert or no-concert days (Trt) and across three time periods (afternoon, 4 to 6 p.m.; evening, 7 to 10 p.m.; night, 10 p.m. to 1 a.m.).

	No Concert			Concert					
	Afternoon	Evening	Night	Afternoon	Evening	Night	Trt	Time	Trt × Time
***Location***									
Box	0.0 ± 0.0	7.2 ± 7.2	7.8 ± 7.8	5.8 ± 5.8	8.6 ± 5.3	10.4 ± 6.8	0.002	0.054	0.171
Den	26.4 ± 16.7 _a_	85.2 ± 6.4 _b_	55.4 ± 18.7 _c_	60.8 ± 15.3 _c_	61.6 ± 9.8 _b_	48.7 ± 12.7 _c_	<0.001	<0.001	0.007
Exhibit front	2.7 ± 1.8	4.2 ± 1.7	20.1 ± 16.1	3.9 ± 2.5	4.3 ± 1.7	11.1 ± 6.1	0.032	0.162	0.334
Exhibit back	4.2 ± 2.6 _a_	3.3 ± 1.9 _a_	0.0 ± 0.0 _a_	4.5 ± 4.5 _a_	12.9 ± 3.9 _b_	4.7 ± 3.7 _a_	0.039	0.012	0.012
***Behaviour***									
Locomotion	0.6 ± 0.4	3.1 ± 0.6	0.8 ± 0.2	0.9 ± 0.7	4.3 ± 1.9	1.6 ± 0.7	0.062	<0.001	0.812
Habitat engagement	5.7 ± 3.9 _a_	11.3 ± 3.4 _b_	2.8 ± 1.4 _c_	2.8 ± 1.0 _c_	10.8 ± 1.9 _b_	4.3 ± 1.7 _a_	0.349	<0.001	0.003
Rest	0.1 ± 0.1 _a_	3.8 ± 3.8 _b_	1.9 ± 1.9 _bcd_	0.6 ± 0.6 _ad_	3.3 ± 2.4 _b_	4.2 ± 3.0 _b e_	0.040	<0.001	0.016
Sleep	0.0 ± 0.0 _a_	0.2 ± 0.2 _a_	5.4 ± 5.4 _b_	6.3 ± 6.3 _b_	4.0 ± 4.0 _b_	3.2 ± 3.0 _b_	<0.001	<0.001	<0.001
Vigilance	0.0 ± 0.0 _a_	3.6 ± 2.1 _b_	0.1 ± 0.1 _a_	0.4 ± 0.3 _ac_	3.3 ± 1.5 _b_	1.6 ± 1.1 _b c_	0.015	<0.001	0.009
Stand	0.0 ± 0.0	0.6 ± 0.2	0.1 ± 0.1	0.3 ± 0.3	0.5 ± 0.2	0.2 ± 0.1	0.826	0.139	0.708
Social	0.0 ± 0.0	0.1 ± 0.1	0.0 ± 0.0	0.0 ± 0.0	0.5 ± 0.3	0.0 ± 0.0	0.164	0.097	-
***Unknown***									
Unknown	12.6 ± 6.1 _a_	46.6 ± 10.7 _b_	43.9 ± 6.7 _bd_	19.2 ± 7.9 _c_	39.2 ± 11.1 _b d_	37.1 ± 11.6 _d_	0.623	<0.001	<0.001
Camera inactive	11.6 ± 6.4 _a_	22.2 ± 8.6 _b_	12.0 ± 5.2 _a_	15.3 ± 8.7 _a_	26.0 ± 9.9 _b_	23.6 ± 13.2 _b_	<0.001	<0.001	0.002

Different subscript letters denote significant changes between treatment days and time periods (*p* < 0.05). - indicates that the behaviour was too rare for analysis.

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
