# Peer review of "A Preliminary Study Investigating the Impact of Musical Concerts on the Behavior of Captive Fiordland Penguins (Eudyptes pachyrhynchus) and Collared Peccaries (Pecari tajacu)"

_animals, 2020, doi:10.3390/ani10112035_

Round 1

Reviewer 1 Report

In this manuscript, the authors assessed the impact of visitors and anthropogenic noise on behaviour of two captive species. Although the assessment is based only on behavioural observations (it would be advisable to associate the behavioural observations with physiological data, e.g. cortisol and corticosterone), the paper is clearly motivated and written, and the insight reported is a useful contribution for the research into ‘visitor effect’, and consequently for health and welfare of zoo animals. Moreover, the diagrams of the exhibits and the map of the zoo are clear help the reader to understand the configuration of the zoo. The paper is accepted in present form

Author Response

Thank you for taking the time to review this paper, and for providing such positive and insightful feedback for us to consider. Due to the majority of the subject animals being kept in groups, conducting non-invasive cortisol analysis (on faecal cortisol samples) was difficult and may have provided inaccurate physiological data. Nevertheless, we will keep this in mind for future research. We appreciate your thoughtful feedback.   

Reviewer 2 Report

Review of “A preliminary study investigating the impact of musical concerts on the behaviour of captive Fiordland Penguins and Collared Peccaries” by Fanning, Larsen and Taylor. A paper submitted to Animals.

This paper presents a preliminary study investigating the impact of musical concerts on a limited number of Penguins and Peccaries. The topic is relevant as musical concerts at public zoos are common in a number different countries and a study investigating the impact of noise from these concerts on zoo animals is useful information to inform Zoo management practices. The paper is preliminary in nature in that it only investigates a small number of animals over a limited time-frame. The study does not draw any strong conclusions regarding any significant effects of the noise from these concerts on the animals, but the findings are interesting enough to warrant further investigation and particular topics for further investigation are described. The study is well executed and the reporting of the experimental method and data analysis could be useful for other similar studies. I recommend that this paper be accepted for publication subject to the minor revisions suggested below.

Minor comments

The paper makes a number of claims about changes in animal behaviour on “concert days” compared with “no-concert days”. However, the concerts only affected the Zoo between 6 pm and 10:30 pm. The observations ran from 4 pm till 1 am.  This is stated in the paper but not in the abstract. I suggest clarifying this point in the abstract. At the moment it is unclear when the observed changes in behaviour occur.

The introduction clearly explains the need for the work and justifies the study and chosen approach using extensive references to appropriate literature. An appropriate description of the species under investigation is also included.

The schematics are of good quality and clearly convey the enclosure layout.

The phrase “musical amplitudes” or “amplitude” is used throughout the paper to describe the measured sound level. This is incorrect. The correct metric should be referred to instead. Presumably what is meant is the sound pressure level, L. If referring to the maximum sound pressure level use max. It should also be made clear whether any frequency weighting was used in these measurements. Note that a “linear weighting” is usually referred to as a Z-weighting (see International Standard IEC 61672). If the sound pressure level indicates a time-average level then express this level as Leq,30min (replacing “30 min” by whatever time period the average is taken over).

Was the sound level meter calibrated? Please state whether it was. Please also state the accuracy of the sound level meter and whether it is a class 1 or class 2 meter. The mounting of the microphone should also be described (e.g. was it mounted on a tripod 1.2 m above ground level and more than 1 m from any solid surface in a direct line-of-sight with the main noise sources?).

I don’t understand why the penguin pool noise “interfered with [the] sound logging accuracy”. Do you mean that the running water increased the background noise at these locations? I think a clearer explanation is required as to why this data was excluded from the study.

There is an extra ( at the end of line 201 on page 2 which should be removed.  

There is some problem with the formatting at the start of section 2.6. Figure 3 seems to be inserted (again) in the middle of a paragraph.

The method for calculating the noise levels presented in table for needs to be explained. Presumably the levels were reported over some time interval (e.g. 1 minute) and a “mean”, max and min level calculated? This is slightly unusual. Noise levels are normally reported as L50, Lmin and Lmax values.

The discussion section is well written and explanations are justified by appropriate references to the literature. The statement that “sound travels poorly in water” is not quite accurate. Sound will certainly not transmit into water easily from a source in air (particularly if the incidence angle of the sound on the surface of the water is acute). This is due to refraction.  

On line 401 it is implied that shrubs will provide a “sound buffer”. This comment should be removed. Foliage will provide almost no sound attenuation. Sheltering inside a den or behind a significant obstacle may certainly provide sound reduction.

Author Response

Thank you for your thorough and helpful review of this paper. Your recommendations were extremely helpful, especially those in regard to properly presenting sound pressure level data with the correct terminology. Below are our responses to your specific comments. 

Comment 1.

The paper makes a number of claims about changes in animal behaviour on “concert days” compared with “no-concert days”. However, the concerts only affected the Zoo between 6 pm and 10:30 pm. The observations ran from 4 pm till 1 am.  This is stated in the paper but not in the abstract. I suggest clarifying this point in the abstract. At the moment it is unclear when the observed changes in behaviour occur.

Addressed Comment 1.

Thank you for drawing attention to this point. We have edited the abstract to include reference to each of the study time periods, as well as added at what time period behavioural changes were observed. Please see the edited section below:

Animal behavior, visitor density and visitor behavior was monitored pre-concert (afternoons; 16:00-19:00), during the concert (evenings; 19:00-21:00) and post-concert (nights; 21:00-00:00) concerts (penguin n = 7; peccary n = 8 days) and in the same periods on days when there was no concert (penguin n = 8 days; peccary n = 6). Fiordland penguins spent more time surface swimming and diving in the pool on concert afternoons and evenings (all (p < 0.001), and more time in the nest on concert nights (p < 0.001), preened less on concert afternoons and nights (p = 0.019) and engaged with their habitat less on concert evenings and nights (p = 0.002) compared to these periods on days without a concert. The group housed peccaries slept more in the afternoon and evening (p ≤ 0.01) and were more vigilant at night (p = 0.009) on concert days compared to no-concert days. The solitary-housed peccary slept more on concert nights(p = 0.035), rested more frequently across all time periods on concert days (p < 0.001), and used the front of the enclosure more across all concert time periods (p < 0.001) compared to no-concert days.

Comment 2.

The phrase “musical amplitudes” or “amplitude” is used throughout the paper to describe the measured sound level. This is incorrect. The correct metric should be referred to instead. Presumably what is meant is the sound pressure level, L. If referring to the maximum sound pressure level use max. It should also be made clear whether any frequency weighting was used in these measurements. Note that a “linear weighting” is usually referred to as a Z-weighting (see International Standard IEC 61672). If the sound pressure level indicates a time-average level then express this level as Leq,30min (replacing “30 min” by whatever time period the average is taken over).

Addressed Comment 2.

Thank you for pointing out and clarifying this. As suggested, we have added additional information pertaining to the sound logger devices and have changed references of “amplitude” to the correct term of “sound pressure level” from Line 228 to 246, as well as all other instances where “amplitude” was referenced throughout the paper. The results section (3.2) has been modified to include dB in relation to sound pressure level (LA – as loggers used an A-weighted frequency response).

Comment 3.

Was the sound level meter calibrated? Please state whether it was. Please also state the accuracy of the sound level meter and whether it is a class 1 or class 2 meter. The mounting of the microphone should also be described (e.g. was it mounted on a tripod 1.2 m above ground level and more than 1 m from any solid surface in a direct line-of-sight with the main noise sources?).

Addressed Comment 3.

Thank you for highlighting this oversight. We have now added mention of that sound level meter being calibrated prior to the study beginning, as well as indicated that the sound level metre was a Class 2 meter. Additionally, we have included a description of the where the microphones were mounted. These changes can be found in Lines 201 to 212 and are as follows:

Class 2 sound level meters (Centre 323 Data Logger Sound Level Meter, Instrument Choice, Adelaide, Australia) were calibrated and then installed in the penguin and peccary exhibits (Figure 3). The peccary sound level meter was mounted atop the wall (> 1m high) separating the two peccary enclosures, and the penguin sound level metre was mounted on a stand ( > 1m high) on the penguin entrance platform (Figure 3). Sound logger devices measured sound pressure levels (LA), measured in decibels (dB), between 30 and 130 dB, and 20 Hz to 8 kHz, across an A-weighted frequency spectrum every minute. Mean, minimum, and maximum sound pressure levels were then calculated for afternoons, evenings, and nights on concert and no-concert days by averaging the dB level over each respective four-hour time period. The sound logger in the penguin enclosure was placed too close to running water which resulted in a near-constant background noise of 73dB. The data could therefore not be used with confidence and was excluded from the study.

Comment 4.

I don’t understand why the penguin pool noise “interfered with [the] sound logging accuracy”. Do you mean that the running water increased the background noise at these locations? I think a clearer explanation is required as to why this data was excluded from the study.

Addressed Comment 4.

Thank you for allowing us to clarify this point. The penguin sound level meter was unfortunately installed too low on its mounting post, thus background noise from the running water in the penguin exhibit registered at a near-constant level of 73dB on the sound loggers. We have added a brief description as to why this data was excluded in Lines 201 to 212 as follows:

The sound logger in the penguin enclosure was placed too close to running water which resulted in a near-constant background noise of 73dB. The data could therefore not be used with confidence and was excluded from the study.

Comment 5.

There is an extra (at the end of line 201 on page 2 which should be removed.  

Addressed Comment 5.

Thank you for identifying this formatting issue. We have removed the additional line.

Comment 6.

There is some problem with the formatting at the start of section 2.6. Figure 3 seems to be inserted (again) in the middle of a paragraph.

Addressed Comment 6.

Thank you for drawing this to our attention. While we did search for this formatting error, we could not see any abnormalities in Section 2.6 and Figure 3 was positioned as expected. If this problem persists, please advise and we will ensure the formatting is corrected.

Comment 7.

The method for calculating the noise levels presented in table for needs to be explained. Presumably the levels were reported over some time interval (e.g. 1 minute) and a “mean”, max and min level calculated? This is slightly unusual. Noise levels are normally reported as L50, Lmin and Lmax values.

Addressed Comment 7.

We appreciate you pointing out this concern. In the methods, we have added reference to how hourly sound pressure levels were calculated (Line 207 to 210), as follows:

Mean, minimum, and maximum sound pressure levels were calculated for afternoons, evenings, and nights on concert and no-concert days by averaging the dB level over each respective four-hour time period.

Additionally, we have edited the information in Table 4 so as to report the sound logger results more accurately, including LAavg, LAmin, LAmax

Comment 8.

The discussion section is well written and explanations are justified by appropriate references to the literature. The statement that “sound travels poorly in water” is not quite accurate. Sound will certainly not transmit into water easily from a source in air (particularly if the incidence angle of the sound on the surface of the water is acute). This is due to refraction.  

Addressed Comment 8.

Thank you for clarifying this point. We have edited the statement to state, “Sound travels poorly from air to water…”

Comment 9.

On line 401 it is implied that shrubs will provide a “sound buffer”. This comment should be removed. Foliage will provide almost no sound attenuation. Sheltering inside a den or behind a significant obstacle may certainly provide sound reduction.

Addressed Comment 9.

We appreciate this being brought to our attention. Whilst we understand that foliage may not be as significant of a sound buffer as a solid object (such as walls in a den or behinds large obstacles), a number of papers suggest that trees and shrubs can provide a substantial sound buffering effect and thus, in Lines 462 to 466, we have edited the original statement of,

This might be due to the environmental features of the exhibit (e.g. shrubs/dens) providing sound buffers and reducing the impact of auditory stress on the animals.

To now read as follows:

This might be due to the environmental features of the exhibit (e.g. shrubs/dens) providing sound buffers and reducing the impact of auditory stress on the animals. Of note, foliage would not provide as significant of a sound buffering effect when compared to more formidable objects such as walls of a den or larger and more solid objects within the exhibit. 

Reviewer 3 Report

This paper explores effects of evening special events on the behavior of zoo-housed penguins and peccaries. As little research exists looking at how after hours events affect captive animals, this paper represents an important contribution to the literature. Overall the paper is well-written and clear. I do have some concerns regarding the literature review in the Introduction, which I’ve spelled out in that section, and some suggestions to improve sentence clarity.

Introduction

Overall the introduction is well-written. However I do have some concerns/recommendations regarding the background literature used.

First, paragraph 2 discusses the effects of novel events. I’d recommend the authors incorporate a recent study on welfare effects of special events in zoos, which was published earlier this year: Bastian et al. 2020 Zoo Biology, “Effects of a recurring late-night event on the behavior and welfare of a population of zoo-housed gorillas”

I also feel that the paragraph on visitor effects in penguins could be more accurately represented as well as more exhaustive, especially given that there aren’t many studies on this topic. For example, the authors mention the studies on Little penguins by Sherwen et al. 2015 and the follow-up in Chiew et al. 2019 but fail to discuss that the negative effects of visitors on penguin behavior was mitigated by merely moving visitors 2m back from the edge of the pool (returning behavior to levels observed when no guests were present at all) or the reason these effects may have been observed in the first place (threatening, predator-like movements by people at the pool edge). Additionally, the authors cite the Ozella et al. 2015 paper but neglect to bring up that the effect of visitor numbers on pool use in the African penguins disappeared after 2 months, suggesting habituation (as someone who also studies visitor effects in penguins, I would consider the evidence of habituation in Ozella et al. 2015 to be a sign of a neutral effect). Visitor effects research is too nuanced for potential misrepresentation/interpretation.

I would also recommend the authors check out the recent literature on visitor effects in penguins suggesting a neutral or even positive effect of visitors, rather than referring to a review and a 20 year old paper. For example, in the same population of African penguins, Ozella et al. 2017 showed no difference in glucocorticoids based on visitor numbers. Below are the references I recommend incorporating:

-Ozella et al. 2017 General and Comparative Endocrinology, “Effect of weather conditions and presence of visitors on adrenocortical activity in captive African penguins”

-Collins et al. 2016 Journal of Zoo and Aquarium Research, “The effect of the zoo setting on the behavioural diversity of captive gentoo penguins and the implications for their educational potential”

-Saiyed et al. 2019 Animals, “Evaluating the Behavior and Temperament of African penguins in a Non-Contact Animal Encounter Program”

Some minor line suggestions:

Line 57: I would recommend removing the comma after “change” at the start of the line and adding the word “and” instead, but this is a personal preference.

Line 59: Also a personal preference, I think the world “challenging” might be preferable over “difficult” here.

Line 96: Instead of “Another”, I would use “A”. Starting with “Another” makes it sound like Fjordland penguins are also common zoo animals, since you just talked about them in a previous paragraph, but you mentioned that particular species of penguin is actually rare in zoos.

Lines 112-114: Please consider adding this sentence to the end of the previous paragraph rather than having it stand alone.

Materials and Methods

Was there any consideration of looking at differences in behavior on days after concerts? Activity on concert days in the time period before the actual concert happens shouldn’t really look any different from normal days (especially the first day of consecutive nights), but I can imagine days after a concert looking different if the animals were disturbed/kept from getting enough rest because of the event the prior evening. I realize the Saturday observations are after the Friday night events but would be interesting to see data for Sunday as well. I understand if those data are not available based on project design, just something to keep in mind for the future.

Section 2.3.1 - If people are paying close attention to the dates of the study compared to when the penguins were rescued, they should realize that this study was the first time the penguins had experienced the concert series, but I think it would be good to add a statement to the end of the first paragraph of this subsection to make it explicit.

Line 149: I would put here that they share the habitat with other penguins rather than saving it for the next paragraph by modifying the end of the second sentence: “...relocated to an outdoor exhibit at Melbourne Zoo which already housed 25 Little penguins” or something similar. I’d then adjust line 153 to just say “allocated to the little penguins.”

Lines 156-158: I think this sentence could be simply added to the end of the previous paragraph.

Line 170: I am unfamiliar with the term “de-sexed males” and think this may be a local phrase? Perhaps a brief definition could be added in parentheses after?

Section 2.3.2 - Similar to my suggestion for 2.3.1, if the peccaries had been exposed to previous Twilights concert series before this study, I would recommend stating that in this subsection. Given their ages, explicitly stating a range of previous concert series experienced (e.g., “Individual peccaries had experienced between 2-4 previous Twilights concert series over their lifetimes at the zoo.”) would be great.

Figures 2 & 3 - can you switch the order of the figure parts to follow the order of the paper? The penguins were discussed first, then the solitary peccary followed by the peccary group.

Line 233: The semi-colon at the end of this line makes it a little unclear, I’d recommend replacing it with “during” and ending the sentence with “periods” or something similar.

Results

Sections 3.1 and 3.4, I’d recommend moving the information on visitor numbers to the beginning of these sections before stating behavior results, as they may influence the behaviors then reported on. I’d also recommend consistent labeling in each section (it’s “Visitor number at the penguin exhibit” in 3.1 and just “Visitors” in 3.4).

Line 271: Should be “greater”

Line 314: Should it be “inactivated”?

Discussion

Line 377-379: Thank you for mentioning this critical point!

Lines 395-399: I’d also recommend checking out a recent paper on zoo soundscapes that discusses dB levels and potential damage to mammalian hearing structures: Pelletier et al. 2020 Zoo Biology, “Zoo soundscape: Daily variation of high-to-low frequency sounds”

Line 421: I can well imagine animals hiding and vigilant in spaces they recognize as safe, so I’m not sure if I’d draw the conclusion that spending more time in those nest boxes automatically means more sleeping.

Author Response

Thank you for your insightful and helpful review of our paper. We especially appreciated your directing us to modern and relevant literature, which added to our introduction and discussion. 

Comment 1.

First, paragraph 2 discusses the effects of novel events. I’d recommend the authors incorporate a recent study on welfare effects of special events in zoos, which was published earlier this year: Bastian et al. 2020 Zoo Biology, “Effects of a recurring late-night event on the behavior and welfare of a population of zoo-housed gorillas”

Addressed Comment 1.  

Thank you for pointing this out and providing us with the opportunity to cite some modern and relevant literature. We have added the following to paragraph 2 of the introduction.

A recent study by Bastian et al., (2020), which found that a majority of zoo-housed gorillas rested less during a late night zoo event compared to before the event and additionally noted changes in individual gorilla behavioural responses to the event, highlights the need to conduct studies on animal response to zoo event days.

Comment 2.

I also feel that the paragraph on visitor effects in penguins could be more accurately represented as well as more exhaustive, especially given that there aren’t many studies on this topic. For example, the authors mention the studies on Little penguins by Sherwen et al. 2015 and the follow-up in Chiew et al. 2019 but fail to discuss that the negative effects of visitors on penguin behavior was mitigated by merely moving visitors 2m back from the edge of the pool (returning behavior to levels observed when no guests were present at all) or the reason these effects may have been observed in the first place (threatening, predator-like movements by people at the pool edge). Additionally, the authors cite the Ozella et al. 2015 paper but neglect to bring up that the effect of visitor numbers on pool use in the African penguins disappeared after 2 months, suggesting habituation (as someone who also studies visitor effects in penguins, I would consider the evidence of habituation in Ozella et al. 2015 to be a sign of a neutral effect). Visitor effects research is too nuanced for potential misrepresentation/interpretation.

Addressed Comment 2.

Thank you for these insightful suggestions. As advised, we have provided more in-depth information regarding penguins in captivity. We have added the following paragraph in Lines 94 to 101 to address these points:

However, Chiew et al., [17] detailed that captive little penguins’ responses to visitor presence was mitigated by regulating visitor proximity to two meters away from the exhibit fence [17]. Additionally, Ozella et al., (2015) noted that captive African penguins reduced pond use in response to high visitor densities following the opening of the exhibit, though this behavioural change was not observed two months into the study, suggesting habituation to visitor presence. It is also recognized that visitor behaviour, rather than visitor density or proximity to exhibits, can be the cause of behavioural changes [7]. 

Comment 3.

I would also recommend the authors check out the recent literature on visitor effects in penguins suggesting a neutral or even positive effect of visitors, rather than referring to a review and a 20 year old paper. For example, in the same population of African penguins, Ozella et al. 2017 showed no difference in glucocorticoids based on visitor numbers.

Addressed Comment 3.

As suggested, we have read the recommended papers and have added the following to Lines 104 to 107 of our paper to address the relevant literature:

Several studies have also outlined that captive penguin species display stable faecal corticosterone levels [22] and increased behavioural diversity [23] in response to visitor presence, suggesting that visitor presence may be perceived as a neutral or positive experience.

 Comment 3.

Line 59: Also a personal preference, I think the world “challenging” might be preferable over “difficult” here.

Addressed Comment 3.

Thank you for providing an alternative. We have changed “difficult” to “challenging.”

Comment 4.

Line 96: Instead of “Another”, I would use “A”. Starting with “Another” makes it sound like Fjordland penguins are also common zoo animals, since you just talked about them in a previous paragraph, but you mentioned that particular species of penguin is actually rare in zoos.

Addressed Comment 4.

Thank you for this suggestion. As recommended, he have changed “Another” to “A” to prevent confusion.

Comment 5.

Lines 112-114: Please consider adding this sentence to the end of the previous paragraph rather than having it stand alone.

Addressed Comment 5.

Thank you, we have done so.

Comment 6.

Was there any consideration of looking at differences in behavior on days after concerts? Activity on concert days in the time period before the actual concert happens shouldn’t really look any different from normal days (especially the first day of consecutive nights), but I can imagine days after a concert looking different if the animals were disturbed/kept from getting enough rest because of the event the prior evening. I realize the Saturday observations are after the Friday night events but would be interesting to see data for Sunday as well. I understand if those data are not available based on project design, just something to keep in mind for the future.

Addressed Comment 6.

Thank you for providing this insight and helpful suggestions. We did not have the resources or available data to consider these approaches but will consider for future studies.

Comment 7.

Section 2.3.1 - If people are paying close attention to the dates of the study compared to when the penguins were rescued, they should realize that this study was the first time the penguins had experienced the concert series, but I think it would be good to add a statement to the end of the first paragraph of this subsection to make it explicit.

Addressed Comment 7.

We appreciate this suggestion and have added the comment, “The two Fiordland penguins had not experienced a Twilights festival previously," to section 2.3.1.

Comment 8.

Line 149: I would put here that they share the habitat with other penguins rather than saving it for the next paragraph by modifying the end of the second sentence: “...relocated to an outdoor exhibit at Melbourne Zoo which already housed 25 Little penguins” or something similar. I’d then adjust line 153 to just say “allocated to the little penguins.”

Addressed Comment 8.

Thank you for this suggestion. We have made these alterations to section 2.3.1.

Comment 9.

Lines 156-158: I think this sentence could be simply added to the end of the previous paragraph.

Addressed Comment 9.

Thank you. As suggested, we have done so.

Comment 10.

Line 170: I am unfamiliar with the term “de-sexed males” and think this may be a local phrase? Perhaps a brief definition could be added in parentheses after?

Addressed Comment 10.

Thank you for pointing this out. We have changed “de-sexed” to “castrated” for clarity.

Comment 11.

Section 2.3.2 - Similar to my suggestion for 2.3.1, if the peccaries had been exposed to previous Twilights concert series before this study, I would recommend stating that in this subsection. Given their ages, explicitly stating a range of previous concert series experienced (e.g., “Individual peccaries had experienced between 2-4 previous Twilights concert series over their lifetimes at the zoo.”) would be great.

Addressed Comment 11.

Thank you for this suggestion. To section 2.3.2, we have added the comment:

All peccaries had experienced between seven and eleven previous Twilight festivals.

Comment 12.

Line 233: The semi-colon at the end of this line makes it a little unclear, I’d recommend replacing it with “during” and ending the sentence with “periods” or something similar.

Addressed Comment 12.

Thank you for drawing this to our attention. As advised, in Lines 223 to 224, we have changed this sentence to:

Behavioural observations were recorded across three distinctive time periods during the afternoons (16:00 – 19:00), evenings (19:00 – 22:00) and nights (22:00 – 1:00) (Figure 4).

Comment 13.

Sections 3.1 and 3.4, I’d recommend moving the information on visitor numbers to the beginning of these sections before stating behavior results, as they may influence the behaviors then reported on. I’d also recommend consistent labeling in each section (it’s “Visitor number at the penguin exhibit” in 3.1 and just “Visitors” in 3.4).

Addressed Comment 13.

Thank you for this excellent suggestion. We have moved visitor data to the beginning of each result section and changed all subheadings to “Visitor number at…” for consistency.

Comment 14.

Line 271: Should be “greater”

Addressed Comment 14.

Thank you, we have edited this spelling error.

Comment 15.

Lines 395-399: I’d also recommend checking out a recent paper on zoo soundscapes that discusses dB levels and potential damage to mammalian hearing structures: Pelletier et al. 2020 Zoo Biology, “Zoo soundscape: Daily variation of high-to-low frequency sounds”

Addressed Comment 15.

Thank you for this suggestion. We have read this suggested paper and found it very insightful, and have added reference to it in the discussion in Lines 458 to 460.

Comment 16.

Line 421: I can well imagine animals hiding and vigilant in spaces they recognize as safe, so I’m not sure if I’d draw the conclusion that spending more time in those nest boxes automatically means more sleeping

Addressed Comment 16.

Thank you for pointing out this discrepancy. We do agree with your assessment of the use of safe refuge zones and have modified Lines 482 to 486 to address this:

Although it is not possible to determine the causality of increased den use, the peccaries may have utilized the den to avoid disturbance related to concert set-up, due to the den potentially being perceived as a safe refuge zone.